# Chemical-Physical Model of Gaseous Mercury Emissions from the Demolition Waste of an Abandoned Mercury Metallurgical Plant

Rafael Rodríguez [1], Begoña Fernández [2] , Beatriz Malagón [3] and Efrén Garcia-Ordiales [1,*]

1 Department of Mining Exploitation and Prospecting, School of Mining, Energy and Materials Engineering, University of Oviedo, Independencia, 13, 33004 Oviedo, Spain
2 Department of Materials Science and Metallurgical Engineering School of Mining, Energy and Materials Engineering, University of Oviedo, Independencia, 13, 33004 Oviedo, Spain
3 Department of Transport and Project and Processes Technology, Polytechnic School of Mining and Energy Engineering, University of Cantabria, Bulevar Ronda Rufino Peón, 254, 39300 Tanos, Spain
* Correspondence: garciaefren@uniovi.es

**Abstract:** Soils from decommissioned Hg mine sites usually exhibit high levels of total mercury concentration. This work examines the behavior of mercury in the atmosphere on samples of contaminated debris of a demolished metallurgical plant present in La Soterraña mine, Asturias (Spain). Previously, a strong dependence of the Hg gas concentration $C_{max}$ ($ng/m^3$) with the temperature T (K) was determined empirically. Hg gas concentration varied between 6500 $ng/m^3$ at low temperatures, 278 K (5 °C), and up to almost 60,000 $ng/m^3$ when the temperature reaches 303 K (30 °C). Then, two different models were proposed to explain the behavior of the mercury emitted from this source. The first model is based on Arrhenius theory. The gas flux per unit area perpendicular to the flow F ($g/sm^2$) is an exponential function of the apparent activation energy $E_a$ (J/mol): $F = c_f \exp(-E_a/RT)$. The values of $c_f = 1.04 \cdot 10^7$ and $E_a = 48.56$ kJ/mol allows the model to fit well with the field measurements. The second model is based on Fick's laws, and the flux F ($g/sm^2$) can be estimated by $F = (K' M_{Hg} p_v)/RT$ where $K' = 8.49 \cdot 10^{-7}$, $M_{Hg} = 200.56$ g/mol and the partial vapor pressure of gaseous mercury $p_v$ (Pa) can be estimated from the saturation vapor pressure of gaseous mercury $p_v = 0.00196 \cdot p_s$ and the August's law $\log(p_s) = 10.184 - 3210.29/T$. This method is also validated with results measured in situ. Both methods are accurate enough to explain and predict emission rate G (g/s), gas flux F ($g/sm^2$) and maximum Hg gas concentration over the debris $C_{max}$ ($ng/m^3$) as a function the temperature T (K).

**Keywords:** mercury; mobility; debris; modelling

## 1. Introduction

Concerns about mercury as an industrial pollutant have led to increased interest in the detection and regulation of mercury in the environment.

The most widely noted Hg reaction in soils is the reduction of $Hg^{2+}$ to $Hg^0$ [1], which is the first step in the volatilization of mercury to the atmosphere. Although the volatilization of $Hg^0$ from soils has been studied extensively, it still represents a primary source of uncertainty in the global/regional annual budget of nonpoint source mercury transport [1,2].

Once released into the air, mercury ($Hg^0$) can be transported to remote areas by atmospheric cycles due to its long residence time in the atmosphere [3,4]. Both natural and anthropic processes can emit a considerable amount of gaseous Hg to the atmosphere [3]. It has been estimated that from the time of the Industrial Revolution, the average Hg concentration in the troposphere has been elevated by a factor of three [5], and a recent publication estimated that the anthropogenic gaseous Hg emissions were 2190 t in 2000 [6,7].

In recent decades, considerable progress has been made regarding the understanding of Hg emission from Hg-enriched areas [8–10]. The results of these studies have demonstrated that Hg emission rates from Hg-enriched areas are higher than background areas [11] and that the contribution of gaseous Hg into the atmosphere from Hg-enriched soil in the mercuriferous belt has largely been underestimated [8,10].

Soils from decommissioned mercury mine sites usually exhibit high concentrations of mercury as a consequence of inefficient mining operations and the usually null restoration actions undertaken [12]. The Principality of Asturias in the north of Spain is a region with mercury mines dating back to the Roman era [13]. The largest Asturian mercury deposits exploited in the past were the La Soterraña (Lena) and El Terronal (Mieres) mines. These mercury mines were part of a substantial and thriving industry in the 19th and 20th centuries. Although mining ceased in 1974, the geochemical dispersion of mercury from abandoned spoil heaps throughout the area continues to be of concern [14–16].

Previous studies on these mines [17] showed a high concentration of gaseous mercury near polluted soil, reaching these last concentrations up to 2–4 orders of magnitude the national background value. Nevertheless, at present, no studies regarding Hg diffusion from waste produced by the demolition of the metallurgical plant have yet been carried out.

The objective of this work is to study the behavior of mercury in the atmosphere in the decommissioned mining area of La Soterraña in Asturias (Spain). More concretely, the study is focused on the behavior of gaseous mercury emitted from contaminated rubble or debris present in the area as a result of the demolition of a metallurgical facility. The study was carried out in the framework of the SUBproducts4LIFE project, a research project co-funded by the European Union as part of the LIFE program. The project SUBproducts4LIFE aims to employ the concept of a circular economy by repurposing industrial waste (coal ash and gypsum from coal power plants and blast furnace slag and steelmaking slag from steel factories) for the large-scale remediation of contaminated soils and brownfield areas associated with Hg mining [17]. The study was carried out in two steps. First, the mercury concentration in the air was monitored in the central contaminated area and in the surroundings, up to 150 m from the central focus. This allowed for the development of an empirical model described in a previous study, [18]. Secondly, a new chemical-physical model was developed and validated by means of the results measured in situ to explain the emission and diffusion of Hg over a short-distance.

## 2. Materials

### 2.1. Site Description

This study was conducted at La Soterraña mine (Lena), 30 km south of the city of Oviedo (Figure 1). The geology of the mine is a low-temperature hydrothermal epigenetic deposit. The predominant minerals are cinnabar (HgS), realgar ($As_4S_4$), and in smaller proportion orpiment ($As_2S_3$). In addition, there are arsenopyrite (FeAsS), marcasite ($FeS_2$), and pyrite (FeS) hosted in fractured limestones and shales. The gangue is composed of carbonates, quartz, and argillaceous minerals (kaolinite and dickite) [19].

The metallurgical plant buildings were demolished in 1989 and debris from the demolition, which has a high mercury content, remained on site.

### 2.2. Measurement of Gaseous Mercury

Gaseous mercury surveys are conducted using portable analytical devices to measure gaseous mercury in the atmosphere. The analytical device measures the Hg concentrations at fixed time intervals and stores the results in an internal data logger. The geographic position is recorded and stored in a GPS device in order to assign coordinates to the geographic position of the recorded data.

A LUMEX RA-915 Hg analyzer is a portable multifunctional atomic absorption spectrometer (AAS) with Zeeman background correction, which eliminates the effect of interfering impurities. This equipment has an analytical gaseous Hg range of 1–100,000 ng/m$^3$ with a detection limit of 0.1 ng/m$^3$ and an uncertainty of Hg measurement range between

6.3 and 9%. The equipment takes in air flow at a rate of 10 L/min and analyzes the Hg concentration with a frequency of one sample per second; the average value is stored every ten seconds in the internal data logger. The LUMEX RA-915 has been widely used in the scientific literature and by reference organizations to monitor gaseous mercury in different conditions and environments. Together with the Hg analyzer, a GPS Garmin Etrex Touch 35 was used. The GPS was programmed to record positions every second to match the data with the Hg measurements. The LUMEX RA-915 analyzes only gaseous mercury because the Hg particulates are removed by means of a filter at the entrance of the analyzer.

Airborne mercury levels in the area have been recorded in previous studies [18].

Furthermore, the monitoring campaigns were designed to be systematic to obtain the most accurate information possible to define the site and design work protocols. As a result, a route was established that included 22 control points located throughout the area where measurements of Hg-gas concentrations were to be performed (Figure 2).

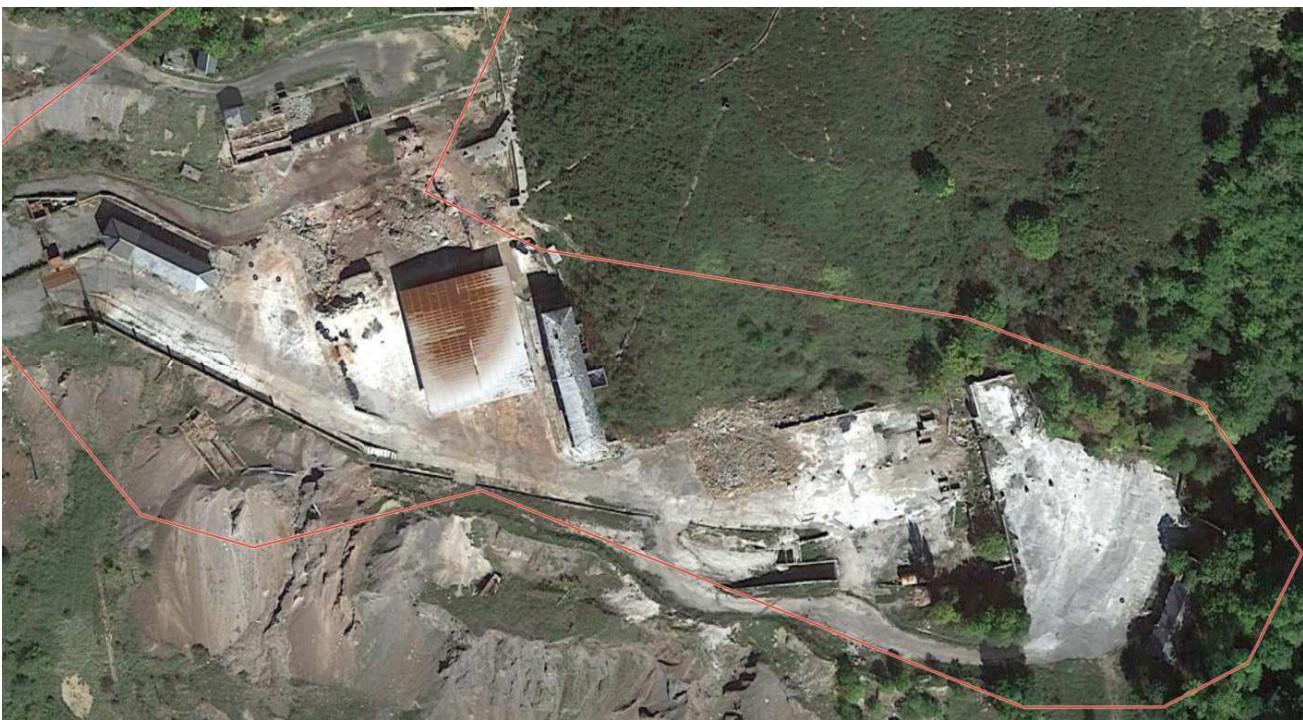

**Figure 1.** Image of the decommissioned mine.

There are three levels or floors: points 1 to 12 are on level 0, points 13 to 17 are on level 1, and points 18 to 22 are on level 2. The height difference between the levels is around 10 m. Levels 0 and 2 are where the work was conducted within the SUBproducts4LIFE project [18,19].

The measurement is made at each place for a time ranging from 2 to 5 min, depending on the observed fluctuations. In the same way, the route will not run more than one hour to reduce exposure.

In this research, a chemical-physical process of a gas which can be influenced by pressure, temperature, or other variables was studied. This made it clear that environmental conditions such as temperature, wind, rain, relative humidity, atmospheric pressure, or solar ultraviolet radiation can influence the emission of gaseous mercury. For example, it can be observed that the wind dilutes the concentration of Hg over the demolition debris. For this reason, measurements were taken in days with low wind speed (average daily speed of 6.5 km/h) and measurements over the debris, 2 to 5 min, were taken when there was no wind. It can also be observed that the emission is lower when it has been raining the previous days due to water evaporation, which can cold the debris and reduce

its temperature. Similarly, clouds can diminish emissions due to lower solar ultraviolet radiation. To avoid the influence of these variables, the measurements were taken on sunny days without rain, clouds, and wind. On the other hand, the range of variation in the atmospheric pressure at this latitude (between 98.0 kPa with low pressure and 104.0 kPa with high pressure) does not significantly influence emission. Finally, although the relative humidity could have an influence, this variable strongly depends on temperature, which means that the gaseous mercury concentration can be expressed as a function of only the temperature. That is, taking the measurements at moments of atmospheric stability with sun, without rain, and without wind, Hg gas emission depends mainly on the temperature, and other conditions produce only an aleatory dispersion of the emission value.

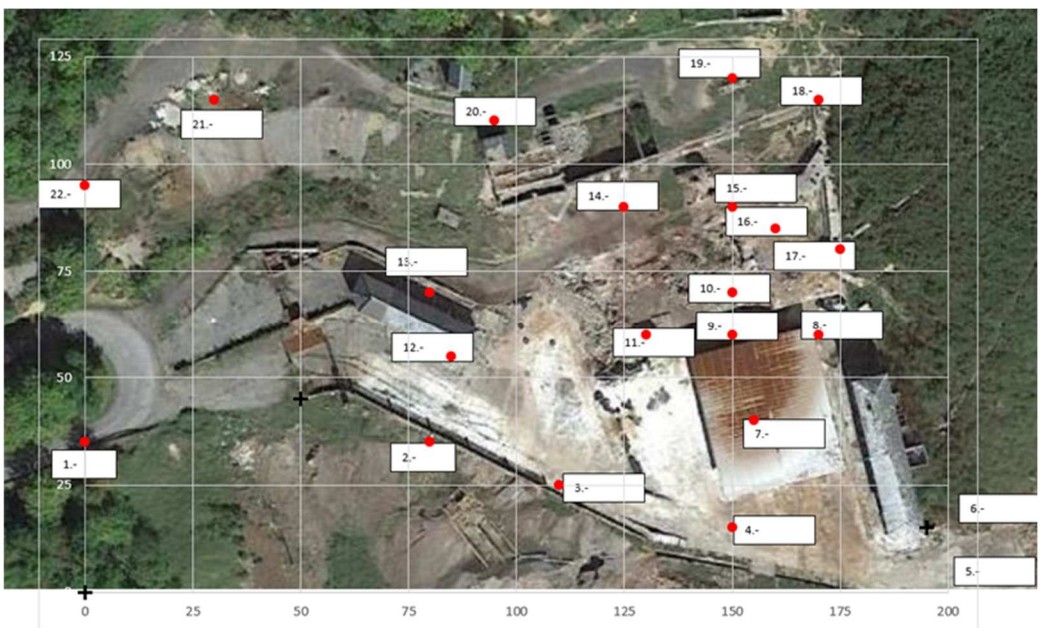

**Figure 2.** A depiction of the sampling route (point 6 is outside of the range of the photograph).

### 2.3. Empirical Models of Emission and Diffusion

The starting points of the present work are the previous empirical models of Hg emission and diffusion defined from Hg concentration measurements.

These measurements were carried out under different temperature conditions (temperatures between 4 °C and 30 °C and without rain or wind) to evaluate the influence of temperature on the potential release of Hg in the area containing demolition debris from the metallurgical plant. In total, four research studies were conducted in the area selected. All the measurements were carried out at 1.0–1.5 m above ground level, which is the recommended height for airborne environmental values [20].

Figure 3 displays these values and demonstrates that the correlation between Temperature (K) and concentration over debris area $C_{10}$ (ng/m$^3$) is explained using Equation (1), [18]:

$$C_{max} = 2.98 \cdot 10^{-5} \, e0.0704 \cdot T \tag{1}$$

A similar relationship was found between the gaseous Hg concentration at point 9, on the border where the debris area ends, $C_9$ (ng/m$^3$) and the temperature (K).

$$C_9 \approx \frac{C_{max}}{2.65} \tag{2}$$

As previously mentioned, measurements were also taken at different distances from the center of the focus. As we move away from the source of contamination, the presence of gaseous mercury can be assimilated to a radial diffusion phenomenon, as can be seen

in Figure 4, which demonstrates how such diffusion follows a hyperbolic function that varies inversely with distance from the source. In Figure 4, dots represent Hg concentration measured under temperatures between 5 °C and 10 °C (A) and between 24 °C and 30 °C (B).

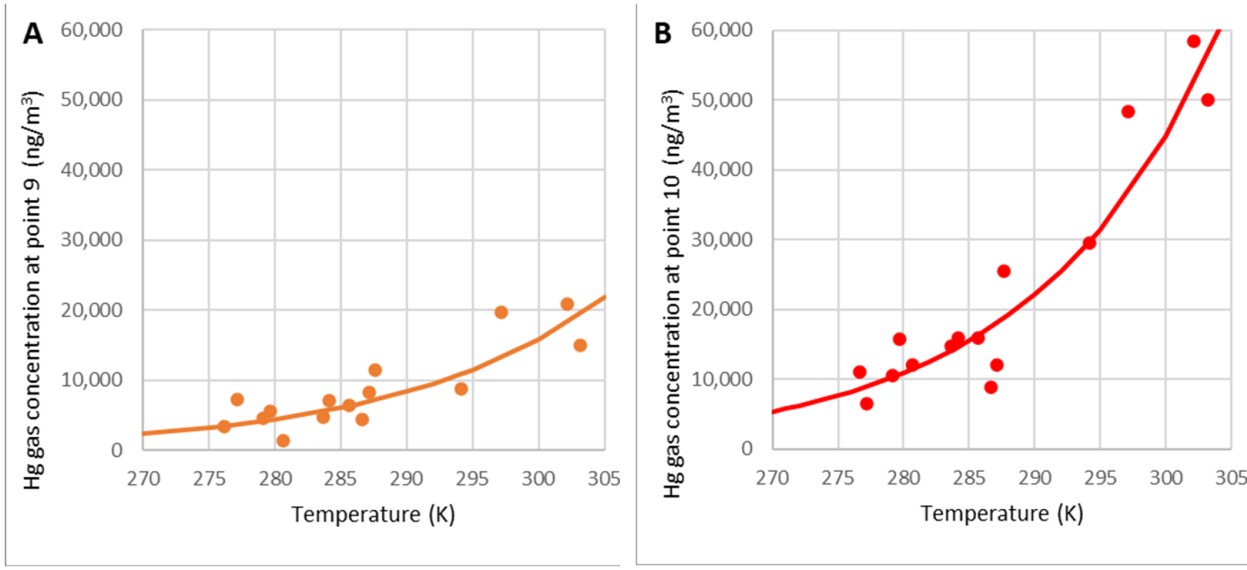

**Figure 3.** Relationship between temperature and concentration at points 9 (**A**) and 10 (**B**).

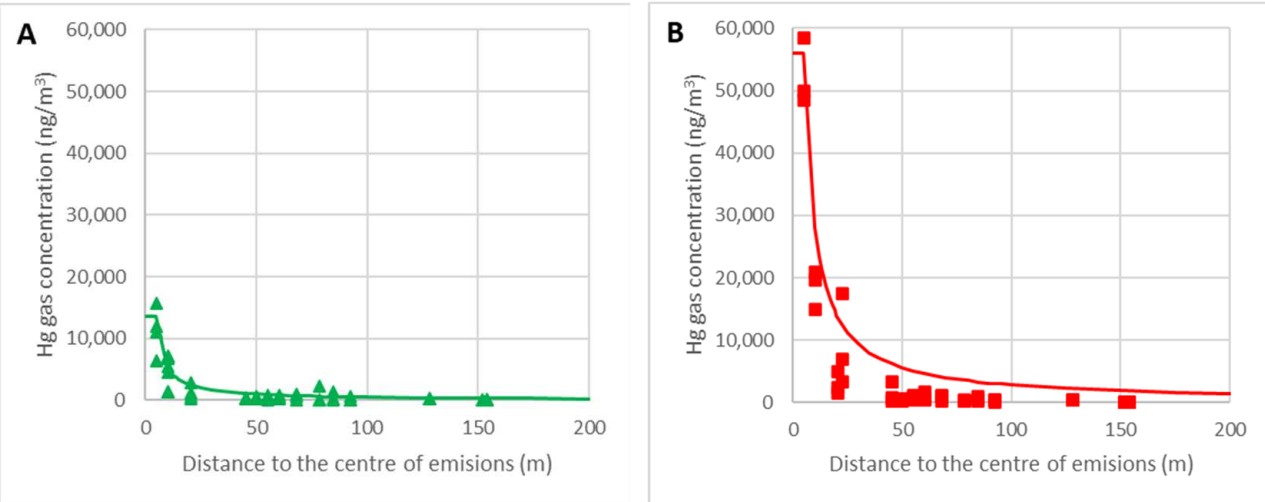

**Figure 4.** Variation in the concentration with the distance from the focus for T = 283 K (**A**) and T = 303 (**B**).

Therefore, the concentration varies with the distance from the focus of the contamination to the measuring point according to the following empirically determined equation:

$$C(r) \approx C_9 \frac{10}{r} \approx \frac{C_{max}}{2.65} \frac{10}{r} \tag{3}$$

The dashed lines in Figure 4 represent the calculated concentration according to Equation (2) for T = 283 K = 10 °C (A) and for T = 303 K = 30 °C (B).

## 3. Methods

### 3.1. Main Hypothesis Regarding Gaseous Hg Flux

In the literature, we found that while the prevalent oxidation states of mercury in the soil are Hg(0), Hg(I) or Hg(II), emission into the atmosphere is predominantly elemental mercury vapor produced by the biotic and abiotic reduction of Hg(II) within the soil [21–23].

Since a detailed model of the reduction of Hg(II) to Hg(0) from debris is not feasible with the information currently available, the following hypotheses were put forward to develop a non-complex model to study the behavior of situations such as the one described above.

The first key point to highlight focuses on the observation of the behavior of waste from the abandoned metallurgical plant, acting as a source of gaseous Hg emissions which can be assimilated to a phenomenon of liquid mercury evaporation.

Following Fick's laws, we can consider two types of flux in the area close to the rubble (Figure 5):

(a)    A flat upward flow diffusion over the rubble area, in which the diffusion velocity decreases with distance from the emission plane.

(b)    A hemispherical radial flow diffusion, outside of the rubble area, in which the concentration will decrease with the inverse of the distance to the center of the rubble area.

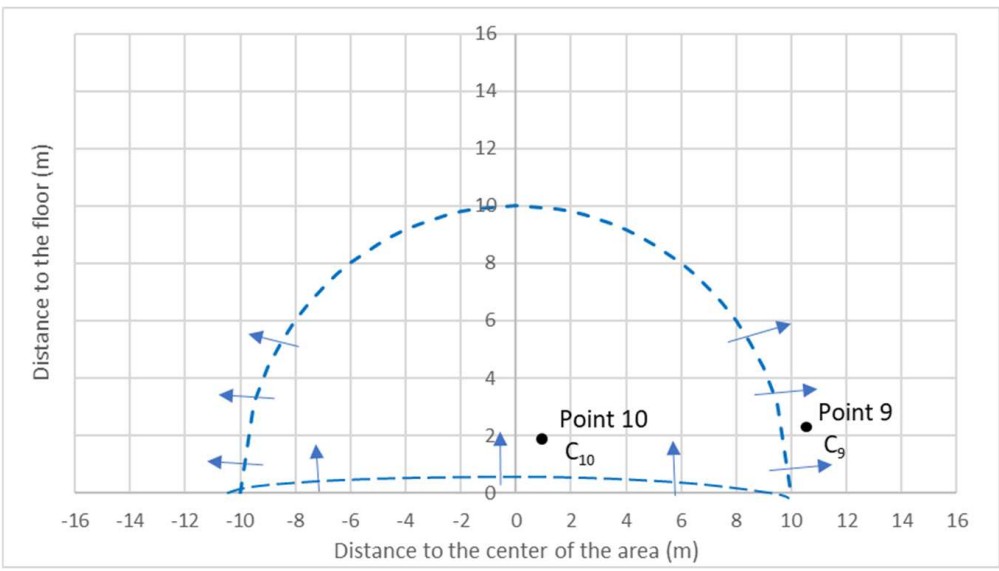

**Figure 5.** Simplified sketch of the gaseous Hg emissions (flat and hemispherical flux).

The procedure was as follows:

(a)    Assuming the radial diffusion, the concentration measurements at the border of the debris area, point 9, makes it possible to determine the emission rate G (ng/s).

(b)    Assuming the flat diffusion near the debris surface, the concentration over the debris, point 10, makes it possible to determine the mass transfer coefficient K' (m/s).

Both parameters allow two models of gaseous Hg emission to be defined: one based on Arrhenius theory like that investigated by [24,25], and another based on liquid evaporation, similar to the development of analytical sensors for the element [26,27].

In this way, the model of emission of the gaseous Hg from the focus and the diffusion of gaseous Hg around the focus is completely defined and can be validated with experimental data.

### 3.2. Models of Radial and Flat Diffusion Based on Fick's Laws

3.2.1. Model of Radial Flow Diffusion and Determination of the Emission Rate G

From the experimental data, it was found that the variation in the gas concentration outside the focus fits a radial flow model. Considering that the gas cloud is above the ground, the flow can be considered hemispheric.

Surrounding the emitting focus of the hemispherical shape of radius $r_0$, the following relationship is fulfilled:

$$y_A = \frac{G_A}{2\pi C D} \frac{1}{r} \tag{4}$$

$y_A$: molar fraction of gas A in the mixture (dimensionless, mol/mol)
$G_A$: emission rate (mol/s).
C: molar concentration of the mixture (mol/m$^3$)
D: diffusivity coefficient (m$^2$/s)
r: distance to the center of the focus (m) at the edge of the debris zone (considered the focus), with $r_9$ = 10 m, the emission ratio $G_A$ for different temperatures can be estimated from the concentrations. Thus,

$$G_A = 2\,\pi\,C\,D\,y_{A9}\,r_9 = 2\,\pi\,C_{A9}\,D\,r_9 \tag{5}$$

where $C_{A9}$ is the molar concentration of the gaseous Hg. By dividing both terms by the molecular mass of mercury ($M_{Hg}$ = 200.59 g/mol), the equation can be re-written:

$$G = 2\,\pi\,D\,C_9\,r_9 \tag{6}$$

where $C_9$ is in g/m$^3$ and G is in g/s (or alternatively ng/m$^3$ and ng/s).

D is the diffusivity coefficient of mercury in the air, which is estimated by applying the formula proposed by [28]:

$$D = D_0 \left(\frac{P_0}{P}\right) \left(\frac{T}{T_0}\right)^{1.81} \tag{7}$$

$D_0$ is the diffusion coefficient of mercury in the air, at standard conditions of pressure and temperature, ($P_0$ = 1 atm = 101.3 kPa and $T_0$ = 293 k), whose value, according to the same author, is $D_0$ = 0.122 cm$^2$ s$^{-1}$ = 1.22 × 10$^{-5}$ m$^2$ s$^{-1}$.

This same expression can be found in the bibliography used by other authors in recent studies related to atmospheric pollution, such as [29,30].

At the limit of the focus, point 9, concentrations of up to 20,000 ng/m$^3$ were measured, similar to those found by [16,31,32] in other similar facilities studied. However, they are much larger than those measured by [33]: around 400 ng/m$^3$, possibly because they were not mineral processing facilities.

On the other hand, if gaseous Hg is produced by the rubble over a floor area A (m$^2$), the flux of gaseous Hg through this area F (ng/s m$^2$) is:

$$F = \frac{G}{A} \tag{8}$$

Table 1 summarizes the calculation of G and F from experimental data T and $C_9$. It is assumed $A \approx \pi\,10^2$ = 314 m$^2$ and $P_0$ = 101,325 Pa.

The G emission rate varies from 0.9 ng/s on the coldest days to 15.5 ng/s on the hottest days. Since the emission surface is assumed to be a circle of a radius $r_9$ = 10 m with an area of 314 m$^2$, the emission flux varies approximately between 10 and 180 ng m$^{-2}$ h$^{-1}$. Therefore, although the measured concentrations are very high, the emission rate is rather low.

The values of F obtained are in the typical range for the different situations, as seen in the review of [34,35], or with those found by [10] at sites contaminated by mercury mining. Nevertheless, they are significantly less than the values reported by [10,36].

On the other hand, there is a dependence of emission flux on temperature, as reported by authors such as [37,38], based on laboratory tests.

**Table 1.** Calculation of emission rate G from experimental data T and $C_9$.

| θ (°C) | T (K) | A (m²) | $C_9$ (ng/m³) | D (m²/s) | G (ng/s) | F (ng/s m²) | F (ng/m² h) |
|--------|-------|--------|---------------|----------|----------|-------------|-------------|
| 29 | 302 | 314 | 20,867 | $1.18 \times 10^{-5}$ | 15.51 | 0.0494 | 177.8 |
| 30 | 303 | 314 | 15,000 | $1.19 \times 10^{-5}$ | 11.22 | 0.0357 | 128.6 |
| 6.5 | 279.5 | 314 | 5524 | $1.03 \times 10^{-5}$ | 3.57 | 0.0114 | 40.9 |
| 7.5 | 280.5 | 314 | 1330 | $1.04 \times 10^{-5}$ | 4.53 | 0.0028 | 9.9 |
| 12.5 | 285.5 | 314 | 6488 | $1.07 \times 10^{-5}$ | 4.36 | 0.0139 | 49.9 |
| 10.5 | 283.5 | 314 | 4689 | $1.06 \times 10^{-5}$ | 3.11 | 0.0099 | 35.6 |
| 4 | 277 | 314 | 7205 | $1.01 \times 10^{-5}$ | 4.58 | 0.0146 | 52.5 |
| 11 | 284 | 314 | 7153 | $1.06 \times 10^{-5}$ | 6.06 | 0.0152 | 54.5 |
| 14 | 287 | 314 | 6426 | $1.08 \times 10^{-5}$ | 5.62 | 0.0139 | 49.9 |
| 14.5 | 287.5 | 314 | 11,493 | $1.08 \times 10^{-5}$ | 7.82 | 00249 | 89.6 |
| 24 | 297 | 314 | 19,681 | $1.15 \times 10^{-5}$ | 14.19 | 0.0452 | 162.7 |
| 21 | 294 | 314 | 8785 | $1.13 \times 10^{-5}$ | 6.22 | 0.0198 | 71.3 |
| 13.5 | 286.5 | 314 | 4387 | $1.08 \times 10^{-5}$ | 2.96 | 0.0094 | 34.0 |
| 3 | 276 | 314 | 3402 | $1.01 \times 10^{-5}$ | 2.15 | 0.0068 | 24.6 |
| 6 | 279 | 314 | 4538 | $1.03 \times 10^{-5}$ | 2.92 | 0.0093 | 33.5 |

### 3.2.2. Model of Flat Upward Flow Diffusion and Determination of Mass Transfer Coefficient K′

The diffusion velocity $v_A$ is constant at any plane perpendicular to the direction of flow and its value is:

$$v_A = \frac{N_A}{C_A} \qquad (9)$$

$v_A$: diffusion velocity (m/s)

$N_A$: molar flux of gas A per unit area perpendicular to flow (mol/s m²)

$C_A$ molar concentration of gas A (mol/m³), since the velocity in a particular plane at a given height is constant, in that plane there will be a proportionality between the gas flow through it and the gas concentration above it:

$$N_A = v_A C_A = K C_A \qquad (10)$$

K: coefficient of proportionality (m/s)

It will be assumed that this proportionality also exists in the upward plane flow produced by the emission of gas from a solid surface, area A (m²), as is the case in this study, with an emission rate G (ng/s).

The concentration $C_i$ (ng/m³) at a given height above the surface will be proportional to the flux F (ng/s m²) at any time:

$$F = \frac{G}{A} = K' C_i \qquad (11)$$

K′: coefficient of proportionality (m/s)

If the movement of the gas in the plane of the surface was due to the pure diffusion of the gas (e.g., if it were the surface of the liquid), the constant K would be equal to the diffusion velocity, which, in the case of the plane flow, would be $v_A = D/z$, where D is the diffusivity coefficient of the gas and z is the distance from the plane to the focus.

However, the movement of the gas from the debris surface is due to a phenomenon of emission from a solid and is therefore not exclusively due to diffusion. The coefficient of proportionality between concentration and flux, K′, is not equal to the diffusion rate, so it must be estimated. The following relationship is fulfilled at the focus:

$$F = \frac{G}{A} = K' C_{10} \qquad (12)$$

where $C_{10}$ (ng/m$^3$) is the Hg concentration in the air above the focus. The calculation of K′ for each temperature is shown in Table 2:

$$K' = \frac{F}{C_{10}} \qquad (13)$$

**Table 2.** Calculation of coefficient K′ (m/s) from concentration $C_{10}$ measured at the focus.

| θ (°C) | T (K) | $C_{10}$ (ng/m$^3$) | G (ng/s) | F (ng/sm$^2$) | K′ (m/s) |
|---|---|---|---|---|---|
| 29 | 302 | 58,488 | 15.51 | 0.0494 | $8.45 \times 10^{-7}$ |
| 30 | 303 | 50,000 | 11.22 | 0.0357 | $7.14 \times 10^{-7}$ |
| 6.5 | 279.5 | 15,827 | 3.57 | 0.0114 | $7.18 \times 10^{-7}$ |
| 7.5 | 280.5 | 12,028 | 4.53 | 0.0028 | $2.29 \times 10^{-7}$ |
| 12.5 | 285.5 | 16,024 | 4.36 | 0.0139 | $8.66 \times 10^{-7}$ |
| 10.5 | 283.5 | 14,757 | 3.11 | 0.0099 | $6.71 \times 10^{-7}$ |
| 4 | 277 | 6512 | 4.58 | 0.0146 | $2.24 \times 10^{-6}$ |
| 11 | 284 | 15,945 | 6.06 | 0.0152 | $9.50 \times 10^{-7}$ |
| 14 | 287 | 12,089 | 5.62 | 0.0139 | $1.15 \times 10^{-6}$ |
| 14.5 | 287.5 | 25,500 | 7.82 | 0.0249 | $9.76 \times 10^{-7}$ |
| 24 | 297 | 48,397 | 14.19 | 0.0452 | $9.34 \times 10^{-7}$ |
| 21 | 294 | 29,518 | 6.22 | 0.0198 | $6.71 \times 10^{-7}$ |
| 13.5 | 286.5 | 8890 | 2.96 | 0.0094 | $1.06 \times 10^{-6}$ |
| 3.5 | 276.5 | 11,011 | 2.15 | 0.0068 | $6.21 \times 10^{-7}$ |
| 6 | 279 | 10,589 | 2.92 | 0.0093 | $8.79 \times 10^{-7}$ |

The value of the global proportionality coefficient K′ can also be found by adjusting the representation of flux F versus concentration $C_{10}$, for all temperatures (Figure 6):

$$F = 8.49 \cdot 10^{-7}\, C_{10} \qquad (14)$$

resulting in a value of K′ = 8.49·10$^{-7}$ m/s with a high correlation coefficient $R^2$ = 0.96.

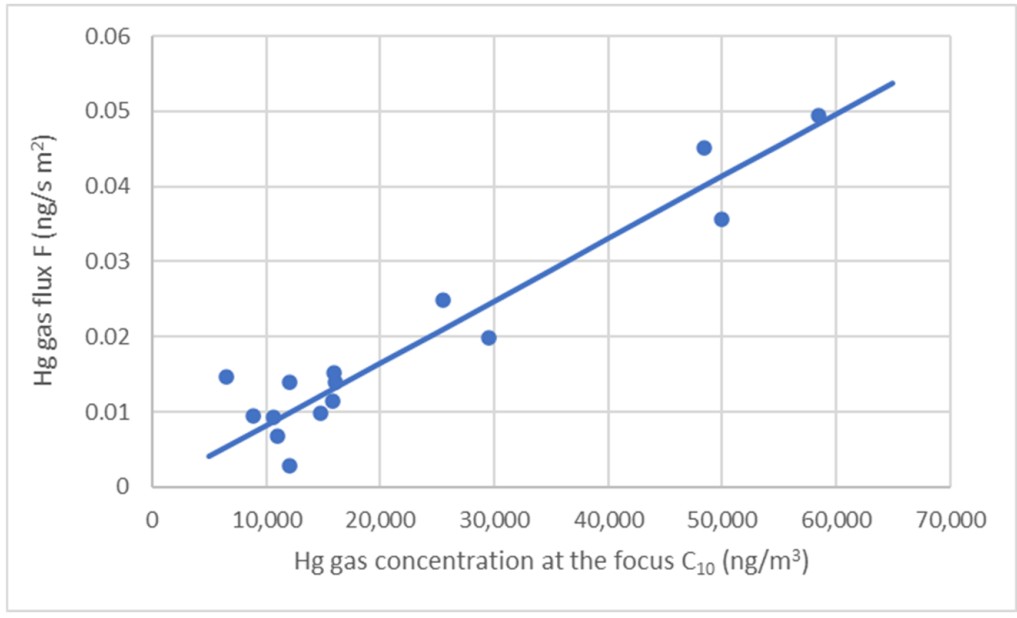

**Figure 6.** Emission flux vs. concentration at focus $C_{10}$.

*3.3. Models of Mercury Emission at the Focus*

Two different models were applied to the focus studied to develop the theoretical model best suited to this type of emission.

### 3.3.1. Model Based on Arrhenius Theory

The more commonly used model in the case of mercury emissions from soils or debris is based on the Arrhenius theory. This approach states that the rate constant k of chemical reactions is related to temperature, as shown in the equation:

$$k(T) = A_f \, \exp\left(-\frac{E_a}{R\,T}\right) \tag{15}$$

where,

k: kinetic constant
$E_a$: activation energy (J/mol)
$A_f$: frequency factor (of the collisions between molecules)
R: ideal gas constant (8.3144 J/mol K = 8.3144 Pa m$^3$ /mol K)
T: temperature (K)

Several authors have used the Arrhenius equation to model the flow of mercury from a solid surface, such as [39–42]. In this case, the relationship is:

$$F = \frac{G}{A} = c_f \, \exp\left(-\frac{E_a}{R\,T}\right) \tag{16}$$

where,

F: gas flux per unit area perpendicular to the flow (g/sm$^2$)
$E_a$: apparent activation energy (J/mol, kcal/mol)
$c_f$: factor related to the Arrhenius $A_f$ constant
G: emission rate G (g/s).
A: area (m$^2$)
R: ideal gas constant (8.3144 J/mol K = 8.3144 Pa m$^3$ /mol K)

Usually, the theoretical activation energy would be equal to the gas evaporation enthalpy. However, since in this case it is not a pure gas evaporation reaction, this activation energy takes a different value and may vary with the types of solids from which the gas is released.

Finally, it should be noted that in no case has UVA solar radiation been evaluated separately. As pointed out by many authors [37,43,44], emissions increase with solar UVA radiation. On cloudy days, emissions would decrease. In this case, the most unfavorable conditions corresponding to the highest emissions were sought out and the measurements were carried out on clear days.

Estimation of the Apparent Activation Energy $E_a$ and the Coefficient $c_f$

The Arrhenius model can be defined for a specific case by determining two parameters, factor $c_f$ and apparent activation energy $E_a$, from the actual measurements taken.

By applying Neper logarithms to the Arrhenius equation, we have a linear relationship between mercury flux F and the variable 1/RT:

$$\ln(F) = \ln\left(\frac{G}{A}\right) = \ln(c_f) - \frac{E_a}{R\,T} \tag{17}$$

Hg concentration measurements $C_9$ made at point 9 have allowed for the determination of the value for mercury emission rate G (ng/s) and mercury flux F (ng/sm$^2$), as summarized in Table 1. When fitting by least squares to a straight line (Figure 7), we obtain the following expression with a correlation coefficient of $R^2 = 0.64$:

$$\ln(F) = 16.16 - \frac{48,562}{R\,T} \tag{18}$$

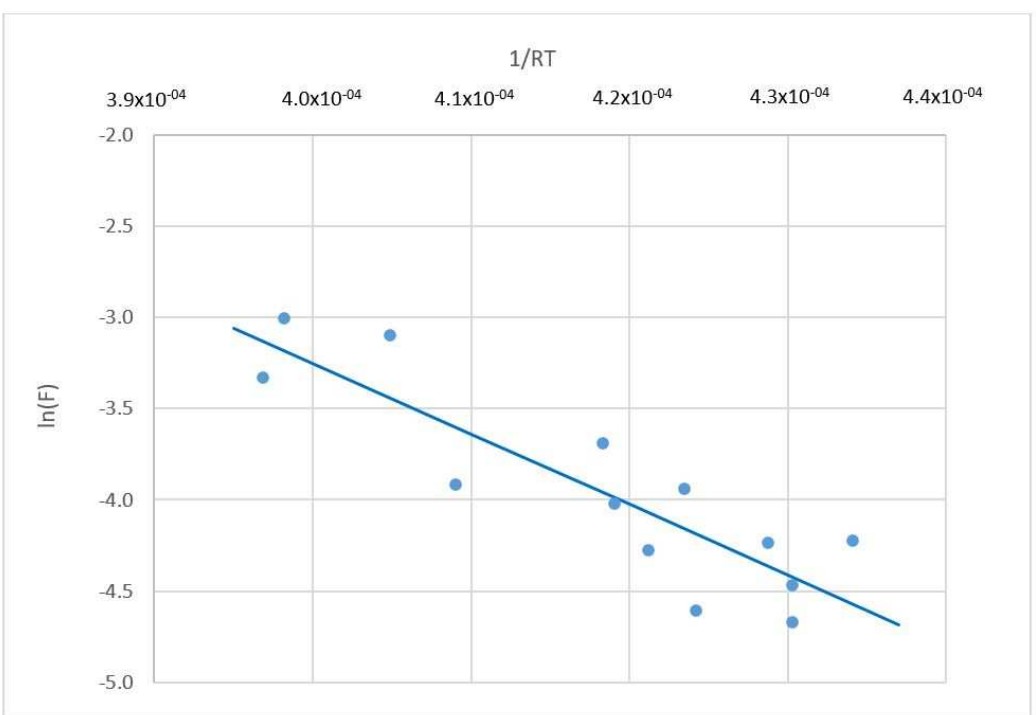

**Figure 7.** Representation of ln(F) versus 1/RT.

From this, it follows that the value of the parameter is $c_f = 1.04 \times 10^7$ and that of the apparent activation energy is $E_a = 48,562$ J/mol = 48.56 kJ/mol. This value of the activation energy, although of the same order, is relatively low compared to the data found in the literature.

For example, the study in [45] found values of 124.8 kJ/mol, the study in [46] provided values between 80 and 110 kJ/mol, the study in [37] reported values between 30 and 75 kJ/mol depending on the origin and concentration of Hg in the soil, and the study in [47] showed values of 131 kJ/mol.

Validation of the Model Based on the Arrhenius Equation

To check the model based on the Arrhenius theory, the model is first used to estimate the gaseous Hg flux and then the results are compared to the experimental data.

In our case, with the previously calculated parameter values: $c_f = 1.04 \times 10^7$; $E_a = 48,562$ J/mol, $A = \pi \times 10^2 = 314$ m$^2$ and $R = 8.3144$ J/K mol, the equation based on the Arrhenius theory is

$$G = A\, c_f\, \exp\left(-\frac{E_a}{R\,T}\right) \Rightarrow G = 3.26{\cdot}10^9 \exp\left(-\frac{5841}{T}\right) \tag{19}$$

which is represented in Figure 8. The G-values thus calculated (dashed line) are plotted together with the G-values estimated from the actual concentration data measured at point 9 (dots). The goodness of fit of the model is evident.

Once the emission rate G is known, the concentration at point 10, $C_{10}$ can be calculated as follows:

$$C_{10} = \frac{G}{A\,K'} \tag{20}$$

Taking into account that the value of the mass transfer coefficient determined previously is $K' = 8.49 \times 10^{-7}$ m/s, the concentration $C_{10}$ can be estimated using the following equation:

$$C_{10} = \frac{c_f}{K'} \exp\left(-\frac{E_a}{R\,T}\right) \Rightarrow C_{10} = 1.22{\cdot}10^{13} \exp\left(-\frac{5841}{T}\right) \tag{21}$$

The mercury concentration at the focus predicted by the model is plotted against the measured data in Figure 9. As can be seen, the model is also useful for the estimation of the concentration of gaseous Hg at the focus.

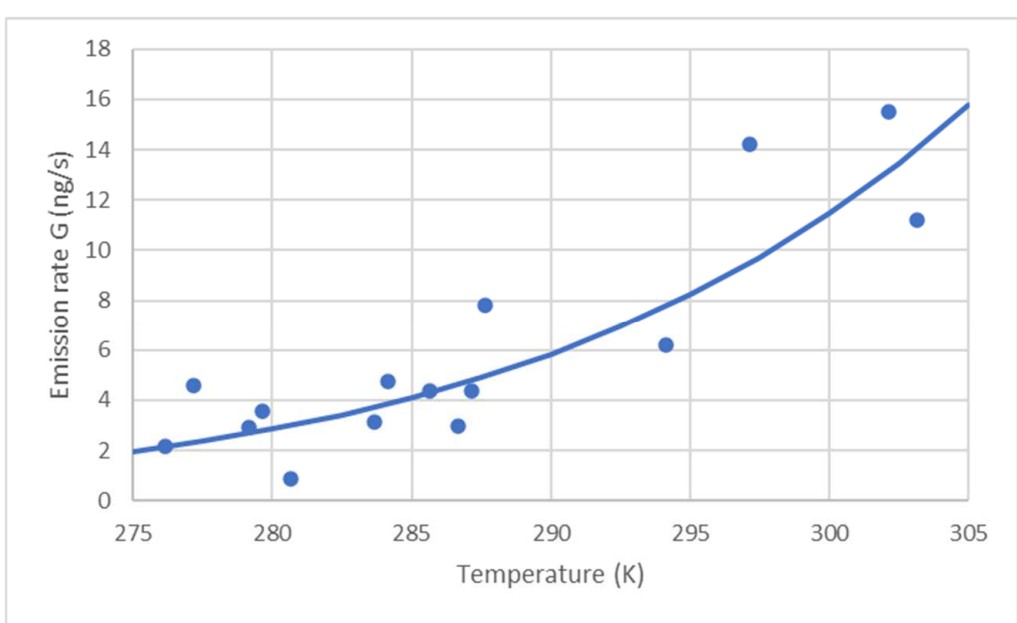

**Figure 8.** Emission Rate vs. temperature values derived from measurements (dots) and calculated with model based on Arrhenius theory (line).

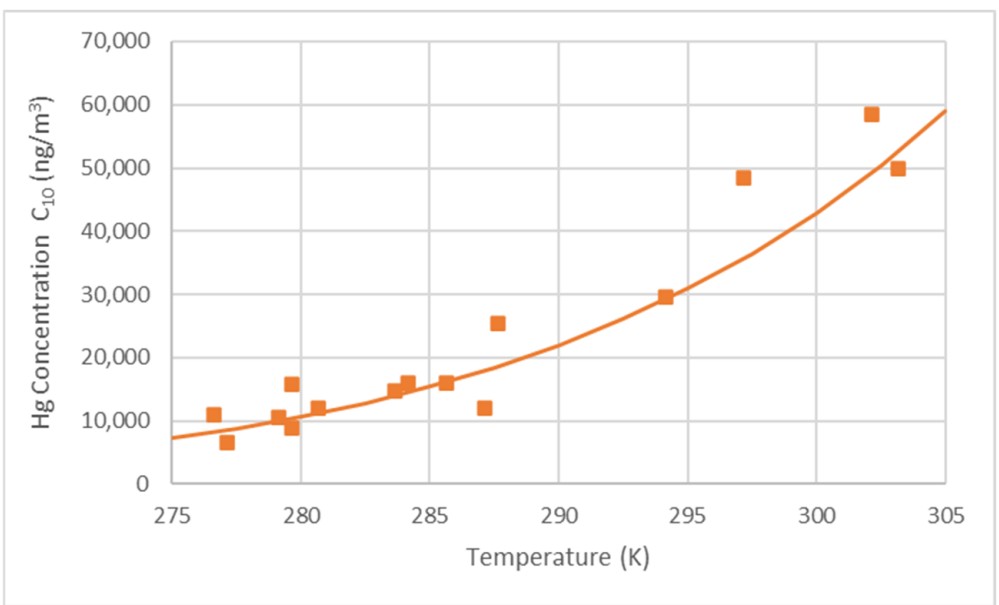

**Figure 9.** Hg concentration $C_{10}$ vs. temperature; measured (dots) and calculated using the model based on Arrhenius theory (line).

### 3.3.2. Model Based on the Laws of Liquid Evaporation

This model was proposed by [48] for the analysis of the formation of pollutant gas clouds (EPA) and is widely accepted; it follows the expression:

$$G = K \frac{A \, M_{Hg} \, p_v}{R \, T} \tag{22}$$

G: emission rate of gases from the liquid to the atmosphere (g/s)

A: liquid surface area ($m^2$)

$p_v$: vapor partial pressure of gaseous mercury in the mixture (Pa)

$M_{Hg}$: molar mass of mercury (g/mol)

T: temperature (K)

R: ideal gas constant (J/mol K = Pa $m^3$ /mol K)

K: mass transfer coefficient (m/s)

The equation is derived from Fick's laws and parameter K is the same as previously explained. In effect, derived from the ideal gas equation, we have for gaseous Hg:

$$C_A = \frac{n}{V} = \frac{p_v}{R\,T} \tag{23}$$

where:

CA: molar mercury concentration (mol/m3)

n: number of moles of gaseous mercury in the mixture (mol)

pv: partial vapor pressure of gaseous mercury in the mixture (Pa)

V: volume of mixture considered ($m^3$)

T: temperature (K)

R: ideal gas constant R = 8.3144 (J/mol K = Pa$m^3$/mol K)

Taking into account

$$M_{Hg}\,N_A = F = M_{Hg}\,K\,C_A \tag{24}$$

From Equations (23) and (24) and rearranging the terms, Equation (22) can be easily deduced.

In this formula, K is the mass transfer coefficient related to the emission of gaseous Hg from the liquid surface due to the evaporation and it can be estimated as described in [48].

In the studied case, it is not a liquid surface, but the surface occupied by a pile of rubble on the ground; therefore, although the equation is the same, the mass transfer coefficient is different i.e., $K'$.

To develop the model, the vapor pressure of mercury gas over the debris, $p_v$, and the transfer coefficient, $K'$, governing the passage of mercury from the debris to the atmosphere, must be determined empirically. Parameter $K'$ has been previously determined and $p_v$ is estimated further on.

Estimation of the Value of Mercury Vapor Pressure $p_v$ over Debris

From measurements of the mercury concentration at the center of focus $C_{10}$, an estimation can be made of the mercury vapor pressure $p_v$ above the debris under working conditions.

As the mercury evaporates and passes into the atmosphere, it diffuses into the gaseous mixture, which is air. Applying the equation of perfect gases to the gaseous mercury in the air above the debris, the vapor pressure will be:

$$p_{v10} = \frac{n\,R\,T}{V} = C_{A10}\,R\,T = C_{10}\frac{R\,T}{M_{Hg}} \tag{25}$$

where $C_{10}$ (g/$m^3$) is the mercury concentration at point 10 over the debris and M is the molecular mass of mercury $M_{Hg}$= 200.59 g/mol.

Table 3 summarizes the calculation of $p_{v10}$ from the data. The value of $p_{v10}$ is low, on the order of $10^{-4}$ Pa in all cases.

Since the temperature range is narrow (0 °C–30 °C or 273 K–303 K), the relationship between vapor pressure and saturation pressure remains almost constant.

In effect, the saturation vapor pressure of gaseous mercury $p_s$ is estimated as a function of temperature, assuming August's law (or a modified Antoine's law):

$$\log_{10}(p_s) = a - \frac{b}{T} \tag{26}$$

$p_s$: saturation vapor pressure of mercury (Pa)

T: absolute temperature in K

a and b are two parameters that must be determined empirically. The values published by [48], a = 10.184 and b= 3210.29 are used here.

Table 3 also shows the values of $p_{s10}$, under these conditions, and the $p_{v10}/p_{s10}$ ratio.

On the other hand, Figure 10 represents $p_{v10}$ versus $p_{s10}$ for the working temperature range. It can be seen as

$$p_{v10} = 0.0196 \, p_{s10} \tag{27}$$

with a correlation coefficient of $R^2 = 0.96$.

**Table 3.** Hg vapor and saturation pressures estimated from the concentration $C_{10}$.

| θ (°C) | T (K) | $C_{10}$ (ng/m$^3$) | $C_{10}$ (mol/m$^3$) | $p_{v10}$ (Pa) | $p_{s10}$ (Pa) | $p_{v10}/p_{s10}$ |
|---|---|---|---|---|---|---|
| 29 | 302 | 58,488 | $2.92 \times 10^{-7}$ | $7.32 \times 10^{-4}$ | $3.55 \times 10^{-1}$ | 0.00206 |
| 30 | 303 | 50,000 | $2.49 \times 10^{-7}$ | $6.28 \times 10^{-4}$ | $3.85 \times 10^{-1}$ | 0.00163 |
| 6.5 | 279.5 | 15,827 | $7.89 \times 10^{-8}$ | $1.83 \times 10^{-4}$ | $4.95 \times 10^{-2}$ | 0.00371 |
| 7.5 | 280.5 | 12,028 | $6.00 \times 10^{-8}$ | $1.40 \times 10^{-4}$ | $5.43 \times 10^{-2}$ | 0.00257 |
| 12.5 | 285.5 | 16,024 | $7.99 \times 10^{-8}$ | $1.90 \times 10^{-4}$ | $8.62 \times 10^{-2}$ | 0.00220 |
| 10.5 | 283.5 | 14,757 | $7.36 \times 10^{-8}$ | $1.73 \times 10^{-4}$ | $7.18 \times 10^{-2}$ | 0.00241 |
| 4 | 277 | 6512 | $3.25 \times 10^{-8}$ | $7.48 \times 10^{-5}$ | $3.89 \times 10^{-2}$ | 0.00192 |
| 11 | 284 | 15,945 | $7.95 \times 10^{-8}$ | $1.88 \times 10^{-4}$ | $7.52 \times 10^{-2}$ | 0.00250 |
| 14 | 287 | 12,089 | $6.03 \times 10^{-8}$ | $1.44 \times 10^{-4}$ | $9.87 \times 10^{-2}$ | 0.00146 |
| 14.5 | 287.5 | 25,500 | $1.27 \times 10^{-7}$ | $3.04 \times 10^{-4}$ | $1.03 \times 10^{-1}$ | 0.00294 |
| 24 | 297 | 48,397 | $2.41 \times 10^{-7}$ | $5.96 \times 10^{-4}$ | $2.35 \times 10^{-1}$ | 0.00254 |
| 21 | 294 | 29,518 | $1.47 \times 10^{-7}$ | $3.60 \times 10^{-4}$ | $1.82 \times 10^{-1}$ | 0.00197 |
| 13.5 | 279.5 | 8890 | $4.43 \times 10^{-8}$ | $1.06 \times 10^{-4}$ | $9.44 \times 10^{-2}$ | 0.00112 |
| 3.5 | 276.5 | 11,011 | $5.49 \times 10^{-8}$ | $1.26 \times 10^{-4}$ | $3.71 \times 10^{-2}$ | 0.00340 |
| 6 | 302 | 10,589 | $5.28 \times 10^{-8}$ | $1.23 \times 10^{-4}$ | $4.95 \times 10^{-2}$ | 0.00248 |

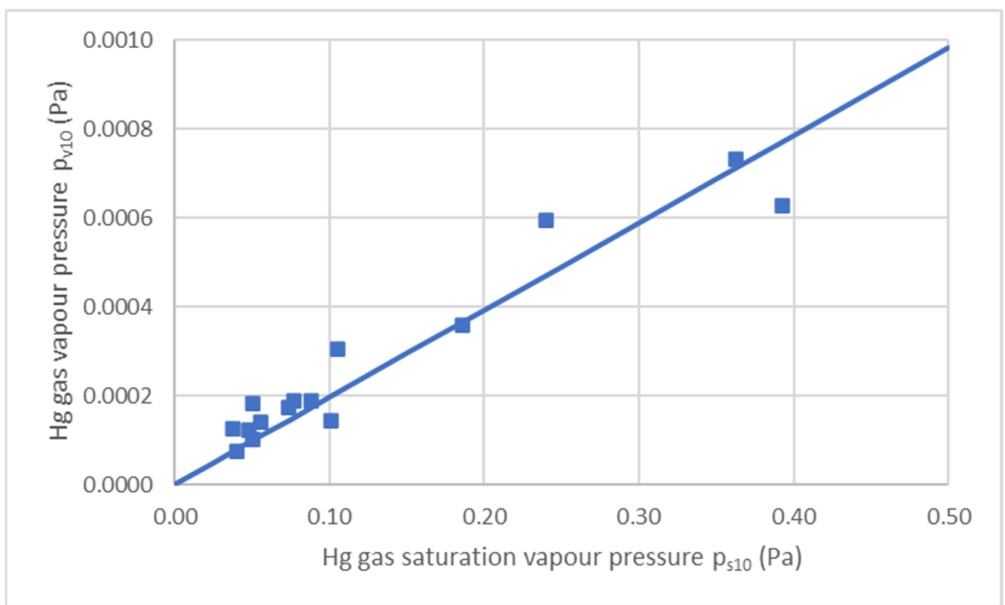

**Figure 10.** Hg vapor pressure $p_{v10}$ versus Hg saturation vapor pressure $p_{s10}$ Validation of the Model Based on the Laws of Liquid Evaporation.

The proposed model is as follows:

$$G = K' \, \frac{A \, M_{Hg} \, p_{v10}}{R \, T} \tag{28}$$

which, in our case, is applied with the following parameter values: $K' = 8.49 \times 10^{-7}$; $A = \pi \times 10^2 = 314 \text{ m}^2$; $M_{Hg} = 200.56 \text{ g/mol}$: $R = 8.3411 \text{ J/mol K}$ and $p_{v10} = 0.00196 \times p_{s10} = 0.00196 \times 10^{(10.184 - 3210.29/T)}$ leaving the expression of G as a function of temperature:

$$G = 1.260 \cdot 10^{-5} \frac{10^{\left(10.184 - \frac{3210.29}{T}\right)}}{T} \tag{29}$$

Figure 11 shows the value of G calculated in this manner along with G values estimated previously from the actual concentration data measured at point 9. It is verified that the model is capable of representing the phenomenon and predicting the emission rate with sufficient accuracy.

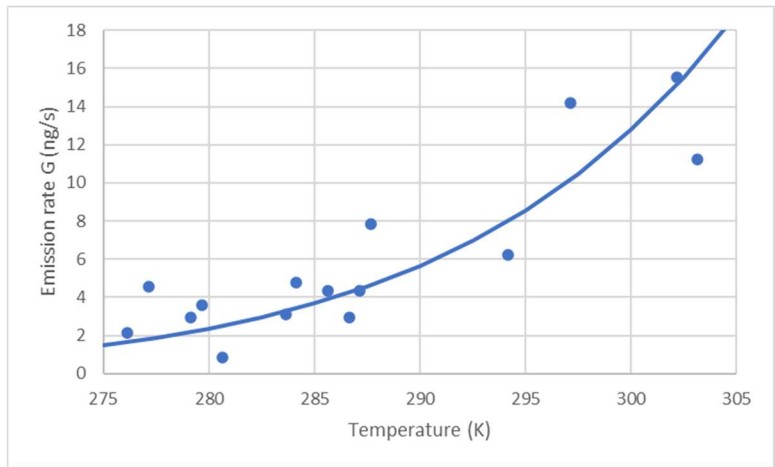

**Figure 11.** Emission Rate vs. temperature values derived from measurements (dots) and calculated with evaporation liquid model (line).

On the other hand, the model can be used to estimate the concentration of gaseous mercury in the air above the debris $C_{10}$:

$$C_{10} = \frac{M_{Hg} \, p_{v10}}{R \, T} \tag{30}$$

As can be seen in Figure 12, the model is also useful for the estimation of the gaseous Hg concentration at the focus.

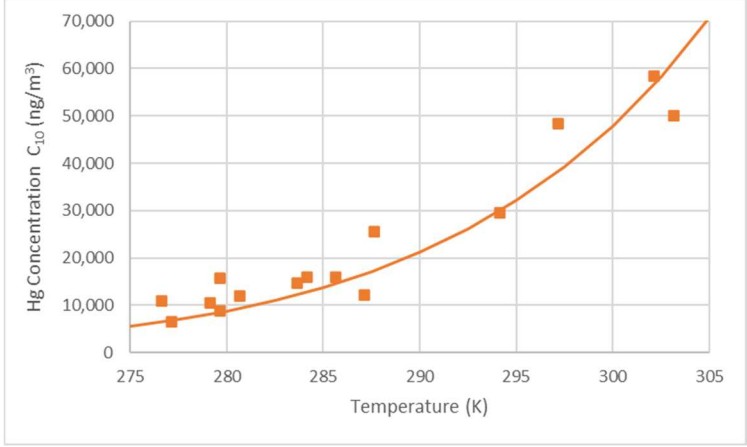

**Figure 12.** Hg concentration $C_{10}$ vs. temperature; measured (dots) and calculated with evaporation liquid model (line).

*3.4. Model of Mercury Dispersion around the Focus*

According to the established hypothesis, the dispersion of gaseous Hg around the focus follows Fick's gas diffusion law, as seen in Equation (4). At any point outside of the debris area, particularly at point 9, the Equation (6) is fulfilled.

By combining both Equations (4) and (6), the following expression can be deduced:

$$C = C_9 \frac{r_9}{r} = C_9 \frac{10}{r} \tag{31}$$

which is the same equation previously determined empirically.

**4. Conclusions**

The behavior of mercury in the atmosphere has been studied from a source of debris from an old dismantled metallurgical facility in a decommissioned mining area in Asturias (Spain).

The study consisted of the monitoring of gaseous mercury concentrations in the contaminated area and developing a model with which to examine the behavior of the mercury emitted by this source.

Two different models were applied to the focus studied to develop the most appropriate theoretical model for this type of emission: one based on the Arrhenius theory and the other on Fick's laws.

In both cases, the methods match actual concentration values at very high levels, although the model based on Fick's laws achieves slightly more favorable levels.

**Author Contributions:** Conceptualization, R.R.; methodology, R.R., B.F., E.G.-O. and B.M.; investigation, R.R. and B.F.; data curation, R.R., B.F. and E.G.-O.; validation R.R., E.G.-O. and B.M.; writing—original draft preparation R.R., and B.F.; writing—review and editing, B.F. and B.M.; supervision: R.R. and B.M. All authors have read and agreed to the published version of the manuscript.

**Funding:** The authors would like to thank the program LIFE of the European Commission for the funding received for the project SUBproducts4LIFE (ref. LIFE16 ENV/ES/000481).

**Institutional Review Board Statement:** Not applicable.

**Informed Consent Statement:** Not applicable.

**Data Availability Statement:** Not applicable.

**Acknowledgments:** Authors would like to thank the collaboration of the institutions and private companies that participated in the project SUBproducts4LIFE: Biosfera consultoría Medioambiental (BIOSFERA), Escorias y Derivados (EDERSA), Global Service (GService), Hidroeléctrica del Cantábrico (EDP), Instituto Asturiano de Prevención de Riesgos Laborales (IAPRL), Recuperación y Renovación (R&R) and Universidad de Oviedo (UNIOVI). Finally, the collaboration of sponsors Arcelor Mittal, Ingeniería de Montajes Norte SA (IMSA), As-turbelga de Minas, and Lena Council, and the Instituto Nacional de Silicosis (INS) is also greatly appreciated.

**Conflicts of Interest:** The authors declare no conflict of interest.

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
