# Peer review of "Chemical-Physical Model of Gaseous Mercury Emissions from the Demolition Waste of an Abandoned Mercury Metallurgical Plant"

_applsci, doi:10.3390/app13053149_

Round 1

Reviewer 1 Report

Post-mining areas affect all natural elements. The article presents the results of mercury concentration measurements in the air at the terain   of a closed mine. The results of measurements were used to perform and validate the model of mercury spreading in the air from mine tailings. The content of the article is interesting and fits the scope of the journal. Below is a list of suggestions for authors:

1. Have Hg2+ concentrations been measured in the air?

2. Please complete the description of the mercury analyzer you are using with the following information:

- What measurement method is used.

- Is the analyzer equipped with a Hg2+/Hg0 converter

- What is the uncertainty of mercury measurement.

3. Does the proposed model take into account the presence of Hg2+ in mercury emitted from the post-mining area? It would be appropriate to write a few words in the introduction about Hg2+ emissions from post-mining areas and how the presence of hg2+ affects the mercury contamination of the environment.

4. Under what conditions are the concentrations given on the graphs?

5. I suggest using the mikro grams/m^3 unit in the charts, it should improve the readability of the charts.

6. Some key data from modeling and mesurements should be added at summary section.

Best regards for authors

Author Response

Reviewer 1

Post-mining areas affect all natural elements. The article presents the results of mercury concentration measurements in the air at the terain   of a closed mine. The results of measurements were used to perform and validate the model of mercury spreading in the air from mine tailings. The content of the article is interesting and fits the scope of the journal. Below is a list of suggestions for authors:

  1. Have Hg2+ concentrations been measured in the air?

Thanks for your comment. Unfortunately, Lumex equipment analizes Total Gaseous Mercury (TGM) without Hg species discrimination.

  1. Please complete the description of the mercury analyzer you are using with the following information:

 - What measurement method is used.

Thank for your comment. This information has been added to the new version of the manuscript.

- Is the analyzer equipped with a Hg2+/Hg0 converter

Thank for your question. As we respond above, the equipment does not discriminate Hg species like other models such as Tekran.

- What is the uncertainty of mercury measurement.

Thank for your comment. This information has been added to the new version of the manuscript.

  1. Does the proposed model take into account the presence of Hg2+ in mercury emitted from the post-mining area? It would be appropriate to write a few words in the introduction about Hg2+ emissions from post-mining areas and how the presence of hg2+ affects the mercury contamination of the environment.

Thanks for your comment. In a previous publication (Rodriguez et. al, 2022), it is noted how mercury gas emission rates in Hg-enriched areas are much higher than the values measured in the background area of non-mining areas. 

However, the empirical model developed in the article predicted that there is no risk of exposure to gaseous mercury from emissions at the mining facility under study. Therefore, the presence of Hg2+ in the atmosphere is not accounted for in the empirical model developed.  

  1. Under what conditions are the concentrations given on the graphs?

Thank you for your comment. The reviewer is right. The explanation about environmental conditions is scarce. Our experience is that taking the measures at moments of atmospheric stability with sun, without rain and without wind, the Hg gas emission depends mainly on the temperature and the other conditions produce only an aleatory dispersion of the emission value. We have added the following paragraph to the text relating to these environmental conditions:

“In this research a chemical-physical process of a gas which can be influenced by the pressure, the temperature, or other variables was studied. In this way it is clear that environmental conditions like temperature, wind, rain, relative humidity, atmospheric pressure or solar ultraviolet radiation can influence the emission of gaseous mercury. For example, it can be observed that the wind dilute the Hg concentration over the demolition debris. For this reason, the measures were taken in days of low wind speed (average daily speed of 6.5 km/h) and the measurement over the debris, 2 to 5 minutes, was taken in a moment of no wind. It can be also observed that the emission is lower when it has been raining the previous days due to the evaporation of water can cold the debris and reduce its temperature. In the same way, clouds make the emissions diminish due to a lower solar ultraviolet radiation. To avoid the influence of these variables, the measures were taken during sunny days without raining, clouds, and wind. On the other hand, the range of variation of the atmospheric pressure at this latitude (between 98.0 kPa with low pressure and 104.0 kPa with high pressure) does not influence the emission significatively. Finally, although the relative humidity could influence, there is a strong dependence of this variable with the temperature which makes that the gaseous mercury concentration can be expressed as a function of only the temperature. This is, taking the measures at moments of atmospheric stability with sun, without rain and without wind, the Hg gas emission depends mainly on the temperature, and the other conditions produce only an aleatory dispersion of the emission value.”

  1. I suggest using the mikro grams/m^3 unit in the charts, it should improve the readability of the charts.

Thank you very much for your suggestion.

The Lumex Hg analyser gives the Hg concentration in nano-grams per cubic meters, and we thought that to use this unit in all cases (formulae, text, graphics…etc.) would be more coherent. In this way we will continue using this unit nano—gram per cubic meter.

Nevertheless, the reviewer is right, and we have included the thousand separator comma to make the graphics more understandable.

  1. Some key data from modeling and mesurements should be added at summary section.

The reviewer is right. Following his suggestion, we have improved the abstract by including the main formulae and the parameter values determined from field data. The new abstract is as following:

“Soils from decommissioned Hg mine sites usually exhibit high levels of total mercury concentration. This work examines the behavior of mercury in the atmosphere on samples of contaminated debris of a demolished metallurgical plant present in La Soterraña mine, Asturias (Spain). Previously, a strong de-pendence of the Hg gas concentration Cmax (ng/m3) with the temperature T (K) was determined empirically. Hg gas concentration varied between 6,500 ng/m3 at low temperatures, 278 K (5º C), up to almost 60,000 ng/m3 when temperature reaches 303 K (30ºC). Then, two different models were proposed to explain the behavior of the mercury emitted from this source.

The first model is based on Arrhenius theory. The gas flux per unit area perpendicular to the flow F (g/sm2) is an exponential function of the apparent activation energy Ea (J/mol): F= cf exp(-Ea/RT). The values of cf= 1.04·107 and Ea= 48.56 kJ/mol makes the model fit well with field measurements. The second model is based on Fick's laws, and the flux F (g/sm2) can be estimated by F=(K’ MHg pv)/RT where K’= 8.49·10-7, MHg=200.56 g/mol and the partial vapor pressure of gaseous mercury pv (Pa) can be estimated from the saturation vapor pressure of gaseous mercury pv= 0.00196·ps and the August's law log(ps)= 10.184 – 3210.29/T. This method is also validated with results measured in situ.

Both methods are accurate enough to explain and predict emission rate G (g/s), gas flux F (g/sm2) and maximum Hg gas concentration over the debris Cmax (ng/m3) as a function the temperature T (K).”

Reviewer 2 Report

Dear Authors,

I found the manuscript entitled “Chemical-physical model of gaseous mercury emissions from the demolition waste of an abandoned mercury metalurgical plant” quite interesting, however, this work necessitates some improvements (listed below) that need to be addressed to better emphasize the contribution of this study to the current knowledge:

1-     Check all manuscript formatting; it is not related to MDPI format, please be consistent, especially in reference format in text and reference in all manuscript. 

2-     Abstract is too short, add some numerical results in it.

3-     Introduction is too long. Please, include more precise and recent literature in the manuscript. Line 77, reference Loredo et al is not completed.

4-     Material and Methods part is very well written.

5-     Results are clear and well written.

6-     Discussion structure seems better but need to support your results in a comparative manner from recent literature. Please edit and enrich discussion with recent literature. 

Author Response

Reviewer 2

Dear Authors,

I found the manuscript entitled “Chemical-physical model of gaseous mercury emissions from the demolition waste of an abandoned mercury metalurgical plant” quite interesting, however, this work necessitates some improvements (listed below) that need to be addressed to better emphasize the contribution of this study to the current knowledge:

 1-     Check all manuscript formatting; it is not related to MDPI format, please be consistent, especially in reference format in text and reference in all manuscript. 

Thanks for this comment. We have we have reviewed the references according to the recommendations of the journal.

2-     Abstract is too short, add some numerical results in it.

The reviewer is right. Following the suggestion, we have improved the abstract by including the main formulae and the parameter values determined from field data. The new abstract is as following:

“Soils from decommissioned Hg mine sites usually exhibit high levels of total mercury concentration. This work examines the behavior of mercury in the atmosphere on samples of contaminated debris of a demolished metallurgical plant present in La Soterraña mine, Asturias (Spain). Previously, a strong de-pendence of the Hg gas concentration Cmax (ng/m3) with the temperature T (K) was determined empirically. Hg gas concentration varied between 6,500 ng/m3 at low temperatures, 278 K (5º C), up to almost 60,000 ng/m3 when temperature reaches 303 K (30ºC). Then, two different models were proposed to explain the behavior of the mercury emitted from this source.

The first model is based on Arrhenius theory. The gas flux per unit area perpendicular to the flow F (g/sm2) is an exponential function of the apparent activation energy Ea (J/mol): F= cf exp(-Ea/RT). The values of cf= 1.04·107 and Ea= 48.56 kJ/mol makes the model fit well with field measurements. The second model is based on Fick's laws, and the flux F (g/sm2) can be estimated by F=(K’ MHg pv)/RT where K’= 8.49·10-7, MHg=200.56 g/mol and the partial vapor pressure of gaseous mercury pv (Pa) can be estimated from the saturation vapor pressure of gaseous mercury pv= 0.00196·ps and the August's law log(ps)= 10.184 – 3210.29/T. This method is also validated with results measured in situ.

Both methods are accurate enough to explain and predict emission rate G (g/s), gas flux F (g/sm2) and maximum Hg gas concentration over the debris Cmax (ng/m3) as a function the temperature T (K).”

3-     Introduction is too long. Please, include more precise and recent literature in the manuscript. Line 77, reference Loredo et al is not completed.Thanks for your comment. We t

Thanks for your comment. In this new version we have tried to reduce its length and added some more recent references to the scientific literature.

4-     Material and Methods part is very well written.

Thanks

5-     Results are clear and well written.

Thanks

 6-     Discussion structure seems better but need to support your results in a comparative manner from recent literature. Please edit and enrich discussion with recent literature. 

Thanks for your comment. We have reviewed this part of the manuscript and we have added new recent references to the text. Also, respect this comment is important remarks that in recent literature it is difficult to find similar to this investigation which highlights this article. Other articles consider other research lines, diffent from those presented in this article in which empirical models validated by real data are proposed. For these reasons, use recent literature is very difficult for us. However, if the reviewer knows of a publication of special interest to improve our research, we would be happy to add your suggestions to improve it.
